# Tissue- and Temporal-Dependent Dynamics of Myeloablation in Response to Gemcitabine Chemotherapy

**DOI:** 10.3390/cells13161317

**Published:** 2024-08-07

**Authors:** Lydia E. Kitelinger, Eric A. Thim, Sarah Y. Zipkowitz, Richard J. Price, Timothy N. J. Bullock

**Affiliations:** 1Department of Pathology, University of Virginia, Charlottesville, VA 22908, USA; lep2gw@virginia.edu (L.E.K.); szipkow1@binghamton.edu (S.Y.Z.); 2Department of Biomedical Engineering, University of Virginia, Charlottesville, VA 22908, USA; eat4nh@virginia.edu

**Keywords:** triple-negative breast cancer, gemcitabine, chemotherapy, tumor microenvironment, tumor immunology

## Abstract

For triple-negative breast cancer (TNBC), the most aggressive subset of breast cancer, immune cell infiltrates have prognostic implications. The presence of myeloid-derived suppressor cells supports tumor progression, while tumor-infiltrating lymphocytes (TILs) correlate with improved survival and responsiveness to immunotherapy. Manipulating the abundance of these populations may enhance tumor immunity. Gemcitabine (GEM), a clinically employed chemotherapeutic, is reported to be systemically myeloablative, and thus it is a potentially useful adjunct therapy for promoting anti-tumor immunity. However, knowledge about the immunological effects of GEM intratumorally is limited. Thus, we directly compared the impact of systemic GEM on immune cell presence and functionality in the tumor microenvironment (TME) to its effects in the periphery. We found that GEM is not myeloablative in the TME; rather, we observed sustained, significant reductions in TILs and dendritic cells—crucial components in initiating an adaptive immune response. We also performed bulk-RNA sequencing to identify immunological alterations transcriptionally induced by GEM. While we found evidence of upregulation in the interferon-gamma (IFN-γ) response pathway, we determined that GEM-mediated growth control is not dependent on IFN-γ. Overall, our findings yield new insights into the tissue- and temporal-dependent immune ablative effects of GEM, contrasting the paradigm that this therapy is specifically myeloablative.

## 1. Introduction

Breast cancer (BrCa) remains the second leading cause of cancer mortality in women. In 2023, about 297,000 new cases of invasive BrCa were diagnosed [1]. This disease is inherently heterogenous. Clinically, the expression of estrogen (ER), progesterone (PR), and human epidermal growth factor 2 (HER2) receptors are used to subdivide BrCa cases—with treatment and prognosis hinging on receptor expression. Triple-negative breast cancer (TNBC) refers to a subgroup of breast tumors lacking ER/PR/HER2 expression. TNBC makes up about 15% of all BrCa cases. Compared to hormone receptor and HER2-positive subtypes, TNBC presents as a more aggressive cancer, with a mortality rate of 40% within 5 years following diagnosis [2,3]. Mortality is often the result of metastatic disease, and patients with TNBC are more likely to experience distant metastatic spread and local recurrence than patients presenting with other BrCa subtypes [4,5,6,7,8]. While historically TNBC has lacked targeted drug and antibody treatments that are available for ER/PR^+^ and HER2^+^ BrCa, in the last few years, pembrolizumab (anti-PD-1; ant-programmed cell death protein 1) immunotherapy has emerged as a promising neoadjuvant and late-stage therapeutic for TNBC patients whose tumors present with a combined positive score of 10 for PD-L1 expression [9,10]. However, only ≈20–34% of TNBC malignancies meet this criterion for pembrolizumab treatment [11,12]. As current research studies aim and continue to identify molecular targets of TNBC [13], cytotoxic chemotherapies remain a systemic treatment option for these patients [14].

Chemotherapy was first used in patients with metastatic BrCa in 1976 when Bonadonna et al. showed that combining cyclophosphamide, methotrexate, and fluorouracil as an adjuvant therapy improved patient outcome [15]. In the decades since, the addition of anthracyclines, taxanes, and various combinations of alkylating agents, antimetabolites, antibiotics, and mitotic spindle inhibitors have been explored as treatment regimens for BrCa both in the adjuvant and neoadjuvant settings [16]. One such chemotherapy is gemcitabine (GEM; 2′2′-difluorodeoxycytidine), a pyrimidine antimetabolite agent that incorporates into the DNA of replicating cells. GEM’s self-potentiating mechanism allows it to be highly effective at inducing cell cycle arrest and apoptosis of proliferating cells [17,18,19]. GEM stands out clinically from other chemotherapies because of its tolerability and low side-effect profile [20]. GEM has shown efficacy as a single agent in BrCa therapy [21,22,23,24,25,26], and its favorable toxicity profile has allowed it to be used in combination with many other antitumor drugs in the context of BrCa [27]. Currently, GEM is being used in the clinic as a third-line treatment for patients with ER^+^ metastatic BrCa, and it is used as a first-line treatment of metastatic TNBC when combined with carboplatin and pembrolizumab [9].

In addition to its nucleoside activity, GEM has been reported to have favorable immunologic effects. BrCa malignancies arise when tumor cells acquire resistance to immunosurveillance and evade elimination by the immune system. This is deemed the “escape” phase of cancer progression, and it is commonly associated with the development of an immunosuppressive tumor microenvironment (TME) [28]. Prominent immune cell subsets involved in pro-tumorigenic immunity are myeloid-derived suppressor cells (MDSCs) [29]. MDSCs—both monocytic (M-MDSCs) and granulocytic (G-MDSCs)—differentiate from immature myeloid cells that undergo high levels of dysregulated myelopoiesis in response to BrCa [30]. Within the 4T1 mammary carcinoma model of TNBC, G-MDSCs are highly prevalent in the TME [31]. MDSCs function as pro-tumorigenic by inhibiting T-cell proliferation and their cytolytic activity [32,33,34], supporting tumor angiogenesis [35,36], triggering metastasis dissemination [37], and inducing the expansion of other suppressive immune cell populations such as regulatory T cells (T_regs_) [38] and tumor-associated macrophages (TAMs) [39]. In preclinical BrCa models, GEM has been deemed a myeloablative therapy, as it significantly reduces the presence of GR-1^+^ MDSCs [40] in the periphery of tumor-bearing mice without perturbing other immune cell populations like CD4^+^ T cells, CD8^+^ T cells, and B cells [41,42]. In the murine 4T1 model, Le et al. showed that GEM acts on suppressive MDSCs that accumulate within mice bearing tumors, and that GEM treatment enhances splenic T-cell expansion and response to antigenic stimuli in vitro [31]. This apparent selective hinderance of MDSCs could be clinically important as MDSC enrichment is associated with poor prognosis in BrCa patients [43]. High levels of circulating MDSCs correlate with liver and bone metastases and higher levels of circulating tumor cells [44], while the presence of tumor-infiltrating lymphocytes (TILs) like CD4^+^ and CD8^+^ T cells in tumors is positively correlated with BrCa prognosis [45,46]. Additionally, TNBC tumors have a greater level of MDSC infiltration than non-TNBC samples [37]. A key motivation of these studies lies in the possibility of selectively targeting MDSCs, increasing TIL presence, and ultimately expanding the number of TNBC patients susceptible to anti-PD-1 immunotherapy. Thus, the repurposing of GEM as a chemo-myeloablative therapy could be an attractive option for augmenting immunotherapies.

While the myeloablative effects of GEM as a treatment for BrCa have been characterized on peripheral immune cells, there is limited understanding of the impacts GEM has on immune populations found in the TME [40,47]. The incomplete knowledge of the effects GEM has on lymphocytes in the TME is especially important given that the presence of TILs is a positive prognostic factor in predicting disease-free survival for TNBC patients [48]. Understanding the intratumoral immunological effects of GEM as a monotherapy is essential when thinking about patient response and potential drug combinatorial therapies against TNBC. We initially hypothesized that GEM would have myeloablative consequences on MDSCs in the TME, while increasing the presence of TILs in TNBC tumors. Utilizing the spontaneously metastasizing murine TNBC model 4T1 [49], we directly compared the impacts of GEM on immune cell presence and functionality in the TME to those found in the periphery. Through bulk-RNA sequencing, we identified immune-related pathways significantly upregulated in TNBC tumors acutely after systemic GEM administration, and we explored whether GEM-mediated constrain of TNBC tumor outgrowth is dependent on the presence of the highly inflammatory cytokine interferon-gamma (IFN-γ). Overall, our findings yield new insights into the tissue- and temporal-dependent immune ablative effects of GEM—a clinically employed chemotherapy. Our findings contrast the current and widely perceived paradigm that this therapeutic is specifically myeloablative.

## 2. Materials and Methods

### 2.1. Cell Line and Animal Maintenance

The 4T1 cell line was maintained in RPMI-1640+L-glutamine (Gibco #11875-093) supplemented with 10% fetal bovine serum (FBS, Corning (Glendale, AZ, USA) #35-010-CV). Cells were grown at 37 °C with 5% CO_2_. Thawed cells were cultured for up to three passages and maintained in a logarithmic growth phase for all experiments. Cells tested negative for mycoplasma prior to freezing.

All mouse experiments were conducted in accordance with the guidelines and regulations of the University of Virginia and approved by the University of Virginia Animal Care and Use Committee. Seven- to nine-week-old female BALB/cJ mice were obtained from The Jackson Laboratory (Jax, Bar Harbor, ME, USA #000651). After shaving, 3.5 × 10^5^ 4T1 cells were subcutaneously (s.c.) implanted into the right flank of mice through a 25G × 1 ½ in needle (BD PrecisionGlide Needle #305127). Mice were housed on a 12 h/12 h light/dark cycle and supplied food ad libitum. Tumor outgrowth was monitored via digital caliper measurements. Tumor volume was calculated as follows: volume = (length × width^2^)/2. Fourteen-days following tumor implantation, mice were randomized into groups in a manner that ensured matching of mean starting tumor volume across experimental groups.

### 2.2. Gemcitabine Therapy

Gemcitabine (GEM; 1.2 mg/mouse in 500 μL volume; Hospira; Kalamazoo, MI, USA) was diluted in 0.9% saline and administered intraperitoneally (i.p.) 14 days after tumor inoculation. For the outgrowth studies, a total of 3 GEM doses were given once a week on days 14, 21, and 28 after the tumor implantations. Mice that did not receive GEM received an i.p. injection of “vehicle” treatment (500 μL of sterile 0.9% saline). GEM dose was based on existing literature demonstrating the use of GEM for inhibition of MDSCs in 4T1 tumor-bearing mice [31].

### 2.3. Flow Cytometry

At 24, 48, and 96 h after GEM administration, tumors and spleens were excised from 4T1 tumor-bearing mice. For blood analysis, mice were bled via tail veins 24, 96, and 168 h following GEM or control injection. Tumors were enzymatically digested for 1 h at 37 °C in RMPI media supplemented with FBS, 20 U/mL Type I Collagenase (Gibco; Grand Island, NY, USA #17018029) and 0.1 mg/mL DNase I (Roche; Branchburg, NJ, USA #10104159001).

After digestion, tumors were subjected to manual homogenization (Wheaton; Ottawa, ON, CAN Tenbroeck Tissue Grinder #62400-518) and filtered through 100 μm filter mesh (Genesee Scientific; Research Triangle Park, NC, USA #57-103) to generate single-cell suspensions, which were then spun down at 1200 RPM for 5 min (Eppendorf 5180; Enfield, CT, USA). Tumor pellets were resuspended in 10 mL of 1× PBS, and then 10 mL of Lympholyte (Cedarlane Labs; Burlington, ON, CAN #CL5035) was underlaid. Tumor samples were centrifuged in the Eppendorf 5180 at 1000× *g* for 20 min with no brake and low acceleration to separate tumor cells from lymphocytes and other immune cells. The layers above the tumor pellet were collected and placed into a clean 50 mL conical tube. 1× PBS was added to fill the conical tube to the 50 mL line. Tumor samples were spun down for 10 min at 800× *g* to pellet cells. These cells were transferred to a 96 well V-bottom plate for staining. Spleens were manually homogenized and filtered through 100 μm filter mesh to generate single-cell suspensions. They were then spun down at 1200 RPM for 5 min and resuspended in 2 mL of Red Blood Cell (RBC) Lysis Buffer (eBioscience; Middletown, VA, USA #00-4333-57) for 2 min. Afterwards, 5 mL of RMPI media containing FBS was added to quench the RBC lysis buffer. Spleens were centrifuged at 1200 RPM for 5 min. Pellets were resuspended in 3 mL of media, and 100 μL of splenocytes were added to a 96 well V-bottom plate for staining. Like spleens, blood samples underwent RBC lysis. Pellets were resuspended in 2 mL of RBC Lysis Buffer for 5 min, quenched with media, and spun down at 1200 RPM for 5 min. Supernatants were decanted, and all the cells were transferred to a 96 well V-bottom plate for staining.

Following an initial wash with 1× PBS, cells were stained for viability using Fixable Live/Dead Blue for 30 min at 4 °C. Next, the samples were exposed to anti-mouse CD16/32 to block Fc gamma receptors for 15 min at 4 °C. Afterwards, cells were washed with FACS buffer; spun down; resuspended in a mixture of Brilliant Stain Buffer and FACS + 2% NMS (normal mouse serum; Valley Biomedical, Inc., Winchester, VA, USA, # AS3054) at a ratio of 1:9, respectively; and stained for 30 min at 4 °C with fluorescent monoclonal antibodies for surface markers: CD45, CD11b, Ly-6G, Ly-6C, F4/80, CD11c, MHCII, Nkp46, CD3, CD4, and CD8α.

Following a wash with FACs buffer, the eBioscience FOXp3/Transcription Factor Staining Buffer Set (#00-5523-00) was used for intranuclear staining. Cells were resuspended in 100 μL of solution 3 parts Fix/Perm Diluent + 1 part Fix/Perm Concentrate for 55 min at 4 °C. Then, the samples were stained for FOXp3, Ki67, IFN-γ, and Arg1 made up in 1× Perm/Wash Buffer at 4 °C for 25 min. Antibody information, supplier name, and catalog number can be found in Appendix A. Following a wash with 1× Perm Buffer, the cells were lastly fixed in 1× BD FACS Lysis in the dark for 10 min at room temperature. They were then resuspended in FACS buffer for running on the cytometer. Flow cytometry was performed with the Cytek Aurora Borealis (Cytek Biosciences; Fremont, CA, USA) and SpectroFlo v3.0.3 software (Cytek Biosciences). Data were analyzed using FlowJo 10 software (FlowJo, LLC; Ashland, OR, USA). tSNE analysis on the granulocytic (Ly6G^+^CD11b^+^) populations in the TME was also performed in FlowJo 10 with an iteration of 1000, perplexity of 30, learning rate (eta) of 10500, KNN algorithm of exact (vantage point tree), and gradient algorithm of Barnes–Hut. All gating strategies can be found in Appendix A.

### 2.4. BFA Injections

To prevent cytokine secretion from cells, 250 µg of Brefeldin A (BFA; Selleck Chemicals; Randor, PA, USA # S7046) diluted in sterilized 0.9% saline was administered intraperitoneally to tumor-bearing mice 4 h prior to harvest.

### 2.5. RNA Extraction, QC, and Library Preparations

At 24 and 168 h (7 days) after GEM administration, tumors were excised from 4T1 tumor-bearing mice and placed into RNAlater solution (Invitrogen; Carlsbad, CA, USA #AM7020). Samples were kept at 4 °C for 24 h and then stored at −80 °C to preserve RNA prior to extraction. The RNeasy Lipid Tissue Mini Kit (Qiagen; Germantown, MD, USA #74804) was used to extract RNA from the tumors. This work was performed by the Biorepository and Tissue Research Facility, which is supported by the University of Virginia School of Medicine, Research Resource Identifiers (RRID): SCR_022971.

RNA QC, library preps, and sequencing were performed by UVA’s Genome Analysis and Technology Core, RRID: SCR_018883. RNA QC was run with the Agilent TapeStation RNA kit. Library preps were performed with the NEBNext^®^ Poly(A) mRNA Magnetic Isolation Module (cat# E7490L) and NEBNext Ultra II Directional RNA Library Prep Kit for Illumina (cat# E7760L). Library QC was performed with the Agilent TapeStation D5000 HS kit and Qubit dsDNA High Sensitivity kit (Cat# Q32851). Sequencing was performed with Illumina’s NextSeq 2000 P3-100 kit.

### 2.6. RNA-Seq Analysis

Quality control

We received 30 million (on an average) paired end reads (on an average 75 bases long read) for each of the replicates sufficient for gene level quantitation. Read quality was assessed using fastqc program http://www.bioinformatics.babraham.ac.uk/projects/fastqc/ (accessed on 31 March 2023), and the raw data quality report was generated using the MultiQC tool [50]. We had a good quality read, and there were no traces of adaptor contamination.

2.Mapping and quantitation

We utilized the “splice aware” aligner “STAR” [51] aligner for mapping the reads. Prior to mapping, we constructed a mouse reference index based on the GRCm38 mouse genome reference (Mus_musculus.GRCm38.dna.primary assembly.fa and Mus_musculus.GRCm38.91.chr.gtf) and setting the “sjdboverhang” parameter to 74 to match the read length of our samples. Subsequently, we conducted read mapping and quantification, and we had more than 95% of the reads mapping to the mouse genome and transcriptome. Gene-based read counts were derived from the aligned reads, and subsequently, a count matrix was generated, serving as the input file for the analysis of differential gene expression.

3.Differential gene expression analysis (DGE):

The DESeq2 package [52] was used to conduct the differential gene expression analysis. Low expressed genes (genes expressed only in a few replicates and had low counts) were excluded from the analysis before identifying differentially expressed genes. Data normalization, dispersion estimates, and model fitting (negative binomial) was carried out with the DESeq function. The log-transformed, normalized gene expression of the 500 most variable genes were used to perform an unsupervised principal component analysis. The differentially expressed genes were ranked based on the log2fold change and FDR corrected *p*-values. Further additional noise was removed using adaptive shrinkage estimators using “apeglm” argument in “lfcShrink” function [53]. The MA plot (function plotMA in DESeq2) and Volcano plot representing significant upregulated and downregulated genes was generated using the specific functions in DESeq2 package. The heat map was generated using the “pheatmap” package in R.

4.Pathway Analysis

Pathway analysis was performed using “fgsea” package in Bioconductor R (https://bioconductor.org/packages/release/bioc/html/fgsea.html, accessed on 31 March 2023). The reference database for mouse pathway enrichment analysis comprised Hallmark, C2, C5, and Hallmark gene sets from the “msigdb” [54,55,56]. For each pathway analysis, we produced a list of the top 10 significant pathways based on *p*-value. Additionally, GSEA-style plots were generated for both upregulated and downregulated pathways.

#### In Vivo Anti-IFN-γ Delivery

To neutralize IFN-γ, 200 µg of either anti-IFN-γ (BioXCell; Bridgeport, NJ, USA XMG1.2 #BE0055) or the IgG1 isotype control (BioXCell HRPN #BE0088) was diluted in sterilized 0.9% saline was administered intraperitoneally to tumor-bearing mice immediately preceding GEM or saline injections on days 14, 21, and 28 after tumor implantation.

### 2.7. Statistical Analysis

Statistical analyses were performed in GraphPad Prism 9 (GraphPad Software). Mouse survival was analyzed using a Kaplan–Meier analysis, and a log-rank (Mantel–Cox) test was used to assess significance. ROUT outliers’ analysis with Q = 0.1% was ran prior to any flow cytometry analysis. Groups of summary data across time were compared using a full-model, two-way analysis of variance (ANOVA) with multiple comparisons, comparing each cell mean with the other cell mean in that row. We acknowledge changes in the immune environment over time after tumor implantation and chose to focus on alterations to the immune repertoire in response to GEM at each individual timepoint, rather than changes over time. When comparing two groups of flow cytometry data at a single timepoint (Appendix A), an unpaired, two-tailed *t*-test with Welch’s correction (i.e., did not assume equal standard deviations) was performed. All figures, unless otherwise stated in the figure legend, show the mean ± standard deviation (SD). *p*-values and significance are specified in figure legends. Graphical abstracts were made with BioRender.com

## 3. Results

### 3.1. Gemcitabine Was Not Acutely Myeloablative in the Tumor Microenvironment

We have previously reported that the combination of GEM and thermally ablative focused ultrasound on 4T1 tumors results in T-cell-dependent control of tumor outgrowth [57]. We hypothesized that GEM augments tumor immunity in this setting, as GEM has been shown to have myeloablative effects on GR-1^+^ MDSCs in the spleens, blood, and bone marrow of 4T1 tumor-bearing mice [31]. To dissect numerical changes in immune cells because of GEM administration to mice bearing 4T1 breast tumors, we utilized spectral flow cytometry to identify cell populations in both the myeloid and T-cell compartments. Because the 4T1 murine TNBC model was deployed for all experiments in this study, it is henceforth referred to as “tumor” or “BrCa”. Consistent with Le et al., when we compared the spleens of GEM treated mice 24, 48, and 96 h post-injection to those of the saline control group, we observed significant reductions in myeloid subsets, including granulocytes (Ly6G^+^CD11b^+^; gating strategy provided in Appendix A), monocytes (Ly6G^−^CD11b^+^F4/80^−^Ly6C^+^), macrophages (Ly6G^−^CD11b^+^F4/80^+^Ly6C^−^), and inflammatory monocytes (Ly6G^−^CD11b^+^F4/80^+^Ly6C^hi^; Appendix A). When enumerating the different myeloid populations, we noticed a specific and drastic reduction in the F4/80^+^Ly6C^hi^ population, which we deemed inflammatory monocytes for their high expression of Ly6C [58]. The effect of GEM in the spleen is acute, as by 96 h, the number of myeloid cells increased, and there was no longer a decrease in the proportion of all live immune cells (Live/CD45^+^) that were granulocytes, monocytes, macrophages, or inflammatory monocytes (Appendix A, respectively). Additionally, we observed the same acute, myeloablative properties of GEM on circulating immune cells when we performed a time course study in tumor-bearing mice. However, by 168 h (7 days) after GEM administration, cell numbers returned to the baseline control levels (gating strategy provided; Appendix A).

Given that the immunosuppressive myeloid cell compartment is significantly diminished systemically within 24 h after GEM treatment, we hypothesized that GEM aids in constraining TNBC outgrowth and significantly lengthens the overall survival time of tumor-bearing mice by depleting Ly6G^+^CD11b^+^ G-MDSCs in the TME (Appendix A). However, when we examined both the number and proportion of granulocytic cells in the TME (Figure 1A; gating strategy provided in Appendix A), we observed no decrease in granulocyte number normalized to the weight of the tumors (Figure 1Ci), or the percentage of all live immune cells that are granulocytes (Figure 1Cii) when comparing the GEM-treated cohort to the saline controls. In fact, the percentage of granulocytic cells in the TME significantly increased 24 and 48 h after GEM treatment (Figure 1Cii), suggesting that the chemotherapy is impacting the presence of other immune cells in the TME. Similar to the granulocytic cells, we found that GEM had no effect on the number of monocytes (Figure 1Di) or the number of macrophages (Figure 1Ei). Although monocytes and macrophages make up a small proportion of the immune cells present in the 4T1 TME, both populations significantly increased 96 h following GEM (Figure 1Dii,Eii, respectively). Unlike the other myeloid subsets, GEM significantly depleted the inflammatory monocyte population within 24 h following administration. However, this effect was acute, as the number of F4/80^+^Ly6C^hi^ cells present in the tumors of GEM-treated mice was comparable to that of the saline-treated mice 48 and 96 h post-injection (Figure 1Fi,Fii).

As an antimetabolite nucleoside analog, GEM acts directly on replicating cells. We predicted that GEM may be preferentially depleting the inflammatory monocytes if the other myeloid subsets are terminally differentiated and the F4/80^+^Ly6C^hi^ cells are more proliferative. However, when we examined proliferation status with Ki67 staining on tumoral myeloid cells, we found that a high proportion (≈80%) of the granulocytic cells in the TME were Ki67^+^ (Appendix A). While the majority of the Ly6G^+^CD11b^+^ cells were Ki67^+^, their overall presence in the TME was not impacted by GEM. Similar findings were observed in the monocyte and macrophage populations as well (Appendix A, respectively). Therefore, cell replication is not the only factor impacting the ablative effects of GEM on immune cell subsets.

### 3.2. GEM Significantly Lowered the Abundance of TILs and Dendritic Cells in the TNBC TME

A risk of administering systemic chemotherapy to patients in the attempt to promote tumor immunity is the non-specific nature of these drugs, and the potential induction of lymphopenia. In the periphery, GEM has been shown to have limited impact on T cell presence [41,42]. Time course analysis of blood and spleens from tumor-bearing mice that either received GEM or vehicle control confirmed the mild effects of GEM on T cell numbers (Appendix A, respectively). In fact, the proportion of total live immune cells that are T-lymphocytes was significantly higher in the GEM-treated cohort (Appendix A). Because the presence of TILs is correlated with positive prognosis for TNBC patients, we investigated the impact of GEM on the presence of T cells within the tumor. Although GEM has a short tissue half-life in mice (≈3 h) [59], we observed a sustained significant reduction in CD8α^+^ T cell presence in GEM-treated tumors compared to the saline controls (Figure 2A,Di,Dii). Although not as striking, CD4^+^ helper T cells (CD4^+^FOXp3^−^) and regulatory T cells (T_regs_; CD4^+^FOXp3^+^) displayed a similar trend in the TME (Figure 2B,Ei,Eii). Additionally, the numbers of intratumoral dendritic cells (DCs; CD11c^+^MHCII^+^), which are essential to provide a secondary co-stimulus for T-cell activation [60], are drastically depleted within these tumors (Figure 2C,Gi). These findings suggest that while GEM displays positive immunological effects systemically, its impact on anti-tumorigenic immune cells may be hindering its ability to elicit a robust immune response against TNBC.

### 3.3. GEM Treatment Acutely Upregulated Genes Associated with Immune Pathways in TNBC Tumors

Given that we observed distinct ablative effects of GEM on immune cells present in the TME, we chose to focus on the TNBC tumors themselves and examined what transcriptional changes are occurring in these tumors as a consequence of GEM treatment. Bulk RNA-seq analysis of 4T1 tumors showed that GEM induces the differential expression of genes associated with immune activation 24 h after administration (Figure 3A). At this timepoint, Gene Ontology (GO) and Gene Set Enrichment Analysis (GSEA) indicated a significant upregulation of the interferon-gamma (IFN-γ response pathway (Figure 3C,E, respectively), which is a hallmark of a type 1 immune response [61]. The strong upregulation of this and other immune pathways such as IFN-α and TNF-α suggests that although GEM is negatively impacting anti-tumoral immune cell populations in the TME, perhaps the cells that remain are being activated. Both the TNF-α and inflammatory response pathways remained upregulated 7 days after GEM injection (Figure 3B,D), indicating that this chemotherapy has prolonged transcriptional effects in these tumors.

### 3.4. Production of IFN-γ, Which Is Predominately Secreted by Granulocytic Cells in the TME, Was Altered by GEM

Because changes at the gene transcriptional level do not always correlate with changes at the protein level, we next determined whether the presence of IFN-γ is altered in the 4T1 TME as a result of GEM treatment. By blocking cytokine secretion in vivo with brefeldin A, we were able to quantify changes in IFN-γ levels at various timepoints following the chemotherapy via flow cytometry (Figure 4A; gating strategy provided in Appendix A). Although there was no difference in the number of total live immune cells in the TME that were producing IFN-γ in the GEM-treated cohort compared to the control mice, there was an acute significant increase in the proportion of live/CD45^+^ cells that were IFN-γ^+^ (Figure 4B,C, respectively). Additionally, we enumerated the amount of IFN-γ being secreted on a per cell basis by analyzing the Geometric Mean Fluorescence (GMF) of IFN-γ on live/CD45^+^ IFN-γ^+^ cells, finding that at 96 h, there was a significant increase in the amount of IFN-γ being produced by these cells isolated from tumors exposed to GEM (Figure 4D).

When looking only at the total live immune cells staining for IFN-γ, we noticed a distinct positive population of cells, so we next determined which cell subset in the TME is responsible for producing IFN-γ. CD8α^+^ cytotoxic [62] and Th1 CD4^+^ T-lymphocytes are known to be a main source of IFN-γ [63]; however, less than 1.5% and less than 0.2% of IFN-γ^+^ cells were CD8α^+^ T cells or CD4^+^ helper T cells, respectively (Figure 4Ei,Fi). While there were no changes in the proportion of CD8α^+^ T cells that were IFN-γ^+^ (Figure 4Eii), there was an increase in the proportion of CD4^+^FOXp3^−^ T cells that expressed IFN-γ 24 h after systemic GEM administration (Figure 4Fii). Only in the CD8α^+^ T cell population was there an increase in the GMF of IFN-γ at the 96 h timepoint (Figure 4Eiii). Surprisingly, the majority (>95%) of IFN-γ^+^ cells were granulocytes (Ly6G^+^CD11b^+^; Figure 4Gi). In both murine and human settings, granulocytic cells have been shown to have the capacity to secrete IFN-γ [64,65,66], and at the acute 24 h timepoint, GEM treatment increased the proportion of IFN-γ^+^ cells that were granulocytes and the proportion of granulocytes that express IFN-γ (Figure 4Gi,Gii, respectively). Similarly, the amount of IFN-γ being secreted on a per cell basis was significantly increased in IFN-γ^+^ granulocytes of the GEM cohort 96 h following injection (Figure 4Giii). We compared these findings in the TME with the effects on GEM on immune cells in the periphery, and we found that GEM significantly decreased both the number and proportion of live/CD45^+^ cells that were IFN-γ^+^, with a trending increase in the GMF of IFN-γ (Appendix A). In the periphery, granulocytes also make up the majority of IFN-γ^+^ cells, so alterations in live/CD45^+^ IFN-γ^+^ cell numbers are likely the result of the myeloablative nature of GEM previously shown in Appendix A (Appendix A). While changes in IFN-γ GMF were only seen at the 96 h timepoint in the TME, IFN-γ^+^ granulocytes in the spleens of GEM-treated tumor-bearing mice displayed elevated IFN-γ GMFs at all three timepoints (Appendix A). Altogether, these findings indicate that systemic treatment with GEM results in changes in the secretion and overall presence of the highly immunomodulatory cytokine, IFN-γ.

### 3.5. GEM-Mediated Tumor Growth Control Was Not Dependent on IFN-γ

We observed that GEM induces alterations to IFN-γ response pathways transcriptionally as well as on the protein level in the TME of TNBC tumors. Next, we examined whether GEM-mediated 4T1 growth inhibition is dependent on IFN-γ. To address this, we co-administered 200 µg of either anti-IFN-γ (BioXCell XMG1.2) or the IgG isotype control with the 3 weekly GEM injections, and tumors were measured daily to assess changes in outgrowth (Figure 5A). We found that neutralizing IFN-γ had no impact on GEM-mediated control of 4T1 outgrowth (Figure 5B,C), suggesting that the anti-tumoral effects of GEM are independent of IFN-γ.

### 3.6. GEM Treatment Altered Arginase 1 Expression by Myeloid Cells in TNBC Tumors

We have shown that GEM does not have a significant impact on the presence of most myeloid cell subsets in the TME, so we investigated whether the suppressive function of these cells is altered by GEM. To achieve this, we quantified changes in Arginase 1 (Arg1) expression. Arg1, a hydrolase enzyme involved in the breakdown of L-arginine, has been shown to accumulate in human and murine breast cancer, and its presence is associated with poor prognosis and an enhanced immunosuppressive environment [67]. To first understand the relatedness of IFN-γ and Arg1 expressing myeloid cells, we performed t-distributed stochastic neighbor embedding (tSNE) dimensionality reduction analysis of flow cytometry data derived from the Ly6G^+^CD11b^+^ population (Figure 6A). Multigraph color mapping of the tSNE plot displays clear separation between the IFN-γ^+^ cells and the Arg1^+^ cells (Figure 6B,C). We concluded that Arg1-expressing cells are discrete from those that express IFN-γ.

We broadly analyzed the expression of Arg1 by total live immune cells in these tumors after treatment with GEM (Figure 6D; gating strategy provided in Appendix A). We observed minimal changes in the number, percentage, and the amount of Arg1 being expressed on a per cell basis within the Live/CD45^+^ population (Figure 6E–G). Because Arg1 is primarily expressed by cells of myeloid lineage in the TME, such as MDSCs and tumor-associated macrophages, we then focused on Arg1 positivity in individual CD11b^+^ cell subsets. We altered the description of the granulocytic and monocytic cells in the TME to be G-MDSCs (Ly6G^+^CD11b^+^Arg1^+^) and M-MDSCs (CD11b^+^ Ly6G^−^F4/80^−^Ly6C^+^Arg1^+^), respectively, because of their pathognomonic expression of immunosuppressive Arg1. G-MDSCs compose the largest subset of Arg1^+^ cells in the TME (70–95%), followed by inflammatory monocytes (5–25%), macrophages (0.53%), and M-MDSCs (0.3-1.5%) (Figure 6H–Kii). The trends within the Arg1^+^ subset mimic the effects of GEM on these immune cell subsets in the TME displayed in Figure 1. GEM did not reduce the number or percentage of G-MDSCs that were Arg1^+^, however, it acutely reduced the presence of Arg1^+^ M-MDSCs and macrophages at 24 h (Figure 6Hi–Ji and 6Hiii–Jiii, respectively). This attenuation was not maintained, as the proportion of MDSCs that were Arg1^+^ returned to baseline by 96 h, while the proportions of Arg1^+^ macrophages and inflammatory monocytes from GEM-treated tumors increased at the later timepoints (Figure 6Jiii, Kiii, respectively). Interestingly, while G-MDSCs comprised the largest subset of Arg1^+^ cells, they had the lowest GMF of Arg1, indicating that these express the lowest levels of Arg1 on a per cell basis when compared to the other three myeloid subsets (Figure 6H–Kiv). Within the M-MDSC and inflammatory monocyte populations, the GMF of Arg1 significantly increased shortly after GEM administration; however, these levels returned to control baseline 96 h after treatment (Figure 6Iiv,Kiv). The opposite trend was observed within the macrophage compartment (Figure 6Jiv). In all, these findings suggest that administering GEM as a therapy for TNBC results in modest changes in the expression of the immunosuppressive enzyme Arg1 in the TME.

## 4. Discussion

Given the prognostic implications of MDSC and TIL presence in the tumors of TNBC patients, the immunosuppressive effects of MDSCs, and the capacity of a subset of TNBC patients to respond to anti-PD-1 treatment, we examined whether the previously reported systemic immunological effects of the clinically relevant chemotherapy, GEM, are recapitulated in the TME of TNBC in a preclinical model. This information is important when considering combination therapies that are likely to improve anti-tumor immunity in TNBC patients with aggressive disease and would thus provide a rationale for the efficacy of GEM in the context of ablative focused ultrasound. Prior research suggests that GEM acts as a myeloablative therapy, depleting suppressive MDSC populations systemically with limited effects on cytotoxic lymphocytes [40,41,42]. Our findings are consistent with the previous study where we examined peripheral CD45^+^ immune cell populations in spleens and blood of 4T1 tumor-bearing mice treated with GEM or vehicle control (Appendix A). However, in contrast to the effects on circulating immune cells, we found that GEM treatment does not have myeloablative effects in the TME; at no assessed timepoint did GEM decrease the granulocyte number in the TME. Rather, the proportion of granulocytes increased in response to treatment. Similar results were seen with the monocyte and macrophage populations (Figure 1).

We further considered the impact of GEM on cellular components that contribute to T-cell responses against tumors. Inconsistent with our hypothesis that GEM would support anti-tumor immunity, we found that contrary to the minimal effects of GEM on T-lymphocyte numbers in circulation after a single dose or multiple doses [41,42,57], there were sustained significant reductions in cytotoxic CD8α^+^ and helper CD4^+^ T cells, as well as a reduction in DCs in the TME following GEM treatment (Figure 2). While immune cells in circulation can feed and supply immune cells to the TME, it is possible that we did not observe the T cells and DCs repopulate over the time course studied because cell extravasation requires proper transmigratory signals such as proinflammatory cytokines, chemoattractant gradients, and adhesive integrins present so that leukocytes can migrate from the bloodstream into the TME [68]. This process can be inhibited by the presence of pro-tumorigenic cytokines and immunosuppressive cells such as MDSCs [69,70]. Nonetheless, although GEM displays positive immunological effects on peripheral immune cell subsets—such as reducing MDSC burden—the diminution of TILs in the TME provides insight as to why GEM as a monotherapy may be unable to induce a robust adaptive immune response and regress TNBC growth (Appendix A), and it suggests that combinatorial regimens utilizing GEM and anti-PD-1 may not be efficacious due to the reduction in TIL presence.

In murine species, GEM has a short half-life of ≈3 h [59], raising the possibility that the ineffectiveness of GEM against MDSCs in the TME could be due to insufficient accumulation in this environment. While we found that the inflammatory monocyte subset of F4/80^+^Ly6C^hi^ cells was significantly reduced acutely following GEM administration (Figure 1), it is possible that sufficient accumulation of the active metabolite of GEM differs amongst immune cell types. We investigated whether GEM was preferentially affecting F4/80^+^Ly6C^hi^ cells due to their proliferative status, as GEM will non-specifically incorporate into the DNA of any replicating cell and induce cell cycle arrest. Surprisingly, we found that all four myeloid populations, including MDSCs, were proliferative in the TME (Appendix A). GEM is a prodrug, meaning it is pharmacologically inactive in its administered state. Functional nucleoside transport proteins are required for this drug to permeate a cell’s plasma membrane [71]. Once inside a cell, GEM must undergo three distinct phosphorylation events to reach its metabolically active form. During its metabolism, GEM can be inactivated at different metabolic stages by deamination via cytidine deaminase or deoxycytidylate deaminase [18,72] and by dephosphorylation by 5′-nucleotidases [73]. Additionally, GEM and its metabolites can also be effluxed from cells by ATP-binding cassette (ABC) transporters [74]. One of the most prominent causes hindering chemotherapeutic efficacy is multidrug resistance. Other mechanisms include increased DNA repair capacity, upregulated anti-apoptotic proteins [75], and activation of the transcription factor NRF2 to support tumorigenesis during oxidative stress [76,77,78]. Therefore, differential protein expression of nucleoside transporters, deaminase enzymes, 5′-nucleotidases, DNA repair proteins, cell death factors, ABC transporters, and/or activation of the NRF2 pathway would impact the efficacy of GEM across immune cell types and could explain the selective ablative effects of GEM. Another possibility is that the penetration of GEM may be locoregional in the TME, resulting in preferential depletion of immune cells residing in areas where GEM may be accumulating. Altogether, our data suggest that DNA replication is not the only factor impacting a cell’s susceptibility to GEM.

Despite the apparent deleterious effects of GEM on the T cell presence in tumors, bulk RNA-seq of TNBC tumors 24 h and 7 days after GEM treatment revealed upregulation of genes and pathways associated with immune activation, one of which being the IFN-γ response pathway. Upon examining IFN-γ production by immune cells in the TME, we discovered that GEM significantly increased the proportion and amount of IFN-γ being secreted by total live immune cells. Interestingly, most of the IFN-γ (>95%) is being produced by granulocytes (Figure 4). While T-lymphocytes and natural killer cells are most often the main producers of this cytokine [62,63,79], neutrophils and granulocytic cells in both the murine and human setting have been shown to produce IFN-γ [64,65,66]. As a potent immunomodulatory cytokine, IFN-γ mediates many tumoricidal mechanisms [80,81,82,83]. However, consistent with previous studies, impeding IFN-γ function did not affect primary 4T1 tumor outgrowth [84,85], and neutralizing IFN-γ did not affect GEM’s ability to constrain 4T1 tumor outgrowth (Figure 5), indicating that the anti-tumoral effects of GEM are not dependent of IFN-γ. These data suggest that the upregulation of IFN-γ is a byproduct of the chemotherapy, by a mechanism yet to be established but possibly related to GEM’s modest ability to induce immunogenic cell death [86]. While other immunosuppressive factors may limit the activity of IFN-γ elicited by GEM in a monotherapy setting, this induction of IFN-γ by GEM may contribute to the enhanced immune responses generated when used in combination with thermally ablative focused ultrasound, per our previous study [57].

The presence of Arg1 in human and murine BrCa correlates with poor prognosis and increased immunosuppression. In the context of BrCa, GEM has been shown to enhance Arg1 expression by Ly6C^hi^ myeloid cells in vitro [47]. Supporting Wu et al.’s findings, we observed a sustained increase in the proportion of inflammatory monocytes (F4/80^+^Ly6C^hi^) that were Arg1^+^ in the GEM treated cohort. Additionally, we found that GEM temporally altered the number of Arg1^+^ cells and the amount of Arg1 being produced on a per cell basis. Reductions in Arg1^+^ cell number are observed in M-MDSC and inflammatory monocyte populations, while we found an increase in Arg1^+^ G-MDSCs 48 h after GEM. Acutely, M-MDSCs and inflammatory monocytes expressed a significant increase in Arg1 GMF, while the macrophage subset had a significant reduction 24 h after GEM administration, followed by a significant increase at 96 h (Figure 5). While further work would need to be done to determine whether GEM is enhancing the immunosuppressive nature of myeloid cells in the TME, our findings support that the chemotherapy is playing a role in augmenting Arg1 expression within these CD11b^+^ cell subsets, suggesting the use of Arg1 inhibitors in combination with GEM.

In conclusion, our data show that treatment of murine TNBC with the clinically employed chemotherapy GEM results in tissue- and temporal-dependent immune ablative effects. While GEM acts as a pan-myeloablative drug in the periphery, it has an isolated impact on F4/80^+^Ly6C^hi^ myeloid cells in the TME. Additionally, we found that a single dose of GEM resulted in a sustained reduction of T-lymphocytes in the TME, potentially hinting at an explanation as to why GEM as a monotherapy is not able to induce a strong adaptive immune response capable of regressing TNBC outgrowth. While we interrogated the immunological impact of a single dose of systemic GEM, future studies could consider whether multiple treatments with this chemotherapy mimic the effects we observe or deviate in terms of immune cell perturbations.

## Figures and Tables

**Figure 1 cells-13-01317-f001:**
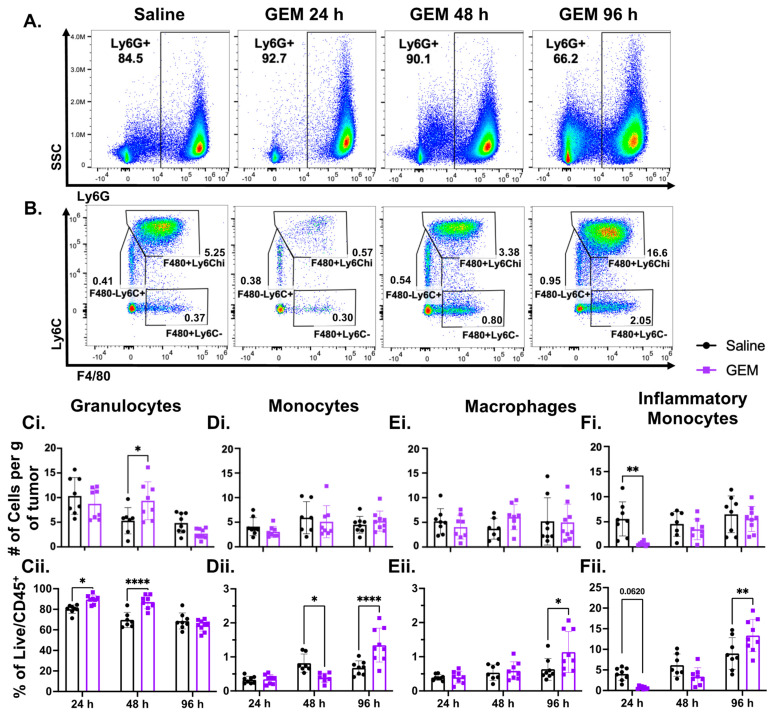
Gemcitabine (GEM) does not act as a myeloablative therapy in the 4T1 tumor microenvironment (TME). 350 k 4T1 parental cells were inoculated in the right flanks of BALB/c mice. Mice were injected with GEM (1.2 mg) or saline I.P. on day 14 post inoculation. Tumors were excised 24, 48 and 96 h post injection (**A**). Representative scatter flow plots showing changes in the granulocyte (CD11b^+^Ly6G^+^) population post GEM administration. (**B**). Representative flow plots of inflammatory monocytes (CD11b^+^Ly6G^−^F4/80^+^Ly6C^hi^), monocytes (CD11b^+^Ly6G^−^F4/80^−^Ly6C^+^) and macrophages (CD11b^+^Ly6G^−^F4/80^+^Ly6C^−^) post GEM injection. All frequencies shown are of the Live/CD45^+^ population. (**C**). Changes in granulocyte number on the order of 10^5^ (**Ci**) and proportion (**Cii**) 24, 48 and 96 h post injection. (**D**). Changes in monocyte number on the order of 10^3^ (**Di**) and proportion (**Dii**). (**E**). Changes in macrophage number on the order of 10^3^ (**Ei**) and proportion (**Eii**). (**F**). Changes in inflammatory monocytes number on the order of 10^4^ (**Fi**) and proportion (**Fii**). (n = 9) (2way ANOVA with multiple comparisons: * *p* < 0.05, ** *p* < 0.01, **** *p* < 0.0001; ROUT Outliers analysis with Q = 0.1%). All points represent mean ± SD.

**Figure 2 cells-13-01317-f002:**
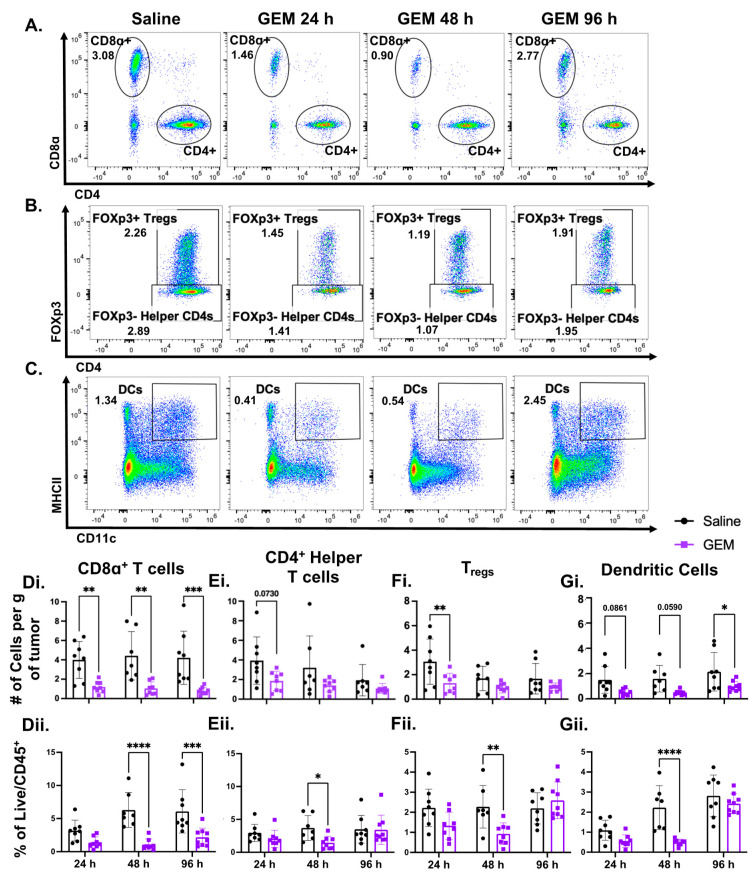
GEM significantly lowers the abundance of tumor infiltrating lymphocytes (TILs) and dendritic cells (DCs) in the TNBC TME. 350 k 4T1 parental cells were inoculated in the right flanks of BALB/c mice. Mice were injected with GEM (1.2 mg) or saline I.P. on day 14 post inoculation. Tumors were excised 24, 48 and 96 h post injection (**A**). Representative flow plots of CD8α^+^ T cells (Ly6G^−^Not DCs CD3^+^CD8α^+^) post GEM injection. (**B**). Representative flow plots of CD4^+^ helper T cells (Ly6G^−^Not DCs CD3^+^CD4^+^FOXp3^−^) and regulatory T cells (T_regs_; Ly6G^−^Not DCs CD3^+^CD4^+^FOXp3^+^). (**C**). Representative flow plots showing changes in dendritic cells (Ly6G^−^CD11c^+^ MHCII^+^) post GEM administration. All frequencies shown are of the Live/CD45^+^ population. (**D**). Changes in CD8α^+^ T cell number on the order of 10^4^ (**Di**) and proportion (**Dii**). (**E**). Changes in CD4^+^ helper T cell number on the order of 10^4^ (**Ei**) and proportion (**Eii**). (**F**). Changes in T_reg_ number on the order of 10^4^ (**Fi**) and proportion (**Fii**). (**G**). Changes in dendritic cell number on the order of 10^4^ (**Gi**) and proportion (**Gii**). (n = 9) (2way ANOVA with multiple comparisons: * *p* < 0.05, ** *p* < 0.01, *** *p* < 0.001, **** *p* < 0.0001; ROUT Outliers analysis with Q = 0.1%). All points represent mean ± SD.

**Figure 3 cells-13-01317-f003:**
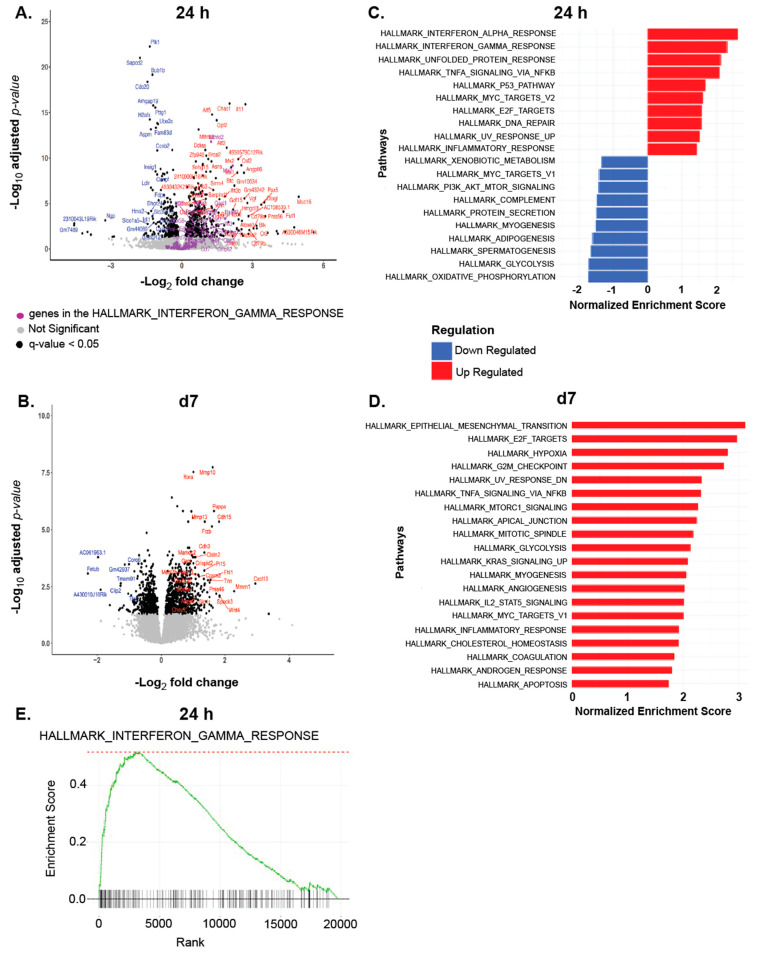
GEM treatment acutely upregulates genes associated with immune pathways in triple negative breast cancer (TNBC) tumors. Bulk RNA-seq was performed on 4T1 tumors 24 h and 7 days post systemic (I.P.) injection of GEM (1.2 mg) or saline vehicle control. (**A**,**B**). Volcano plots of differentially expressed genes in 4T1 tumors from GEM treated and saline treated mice 24 h (**A**) and 7 days (**B**) post injection. (**C**,**D**). Gene ontology analysis showing normalized enrichment scores for the top 20 hallmark pathways modulated in the GEM cohort compared to the control saline cohort 24 h (**C**) and 7 days (**D**) post injection. (**E**). Gene seat enrichment analysis for the “INTERFERON GAMMA RESPONSE” pathway upregulated 24 h after GEM administration.

**Figure 4 cells-13-01317-f004:**
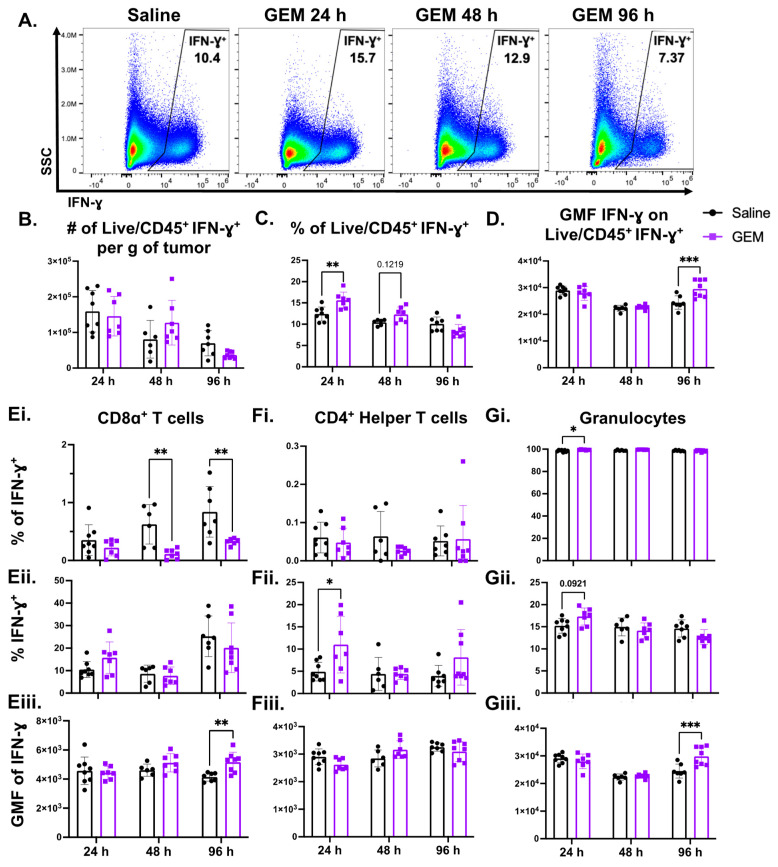
Production of IFN-γ, which is predominately secreted by granulocytic cells in the TME, is altered by GEM. 350 k 4T1 parental cells were inoculated in the right flanks of BALB/c mice. Mice were injected with GEM (1.2 mg) or saline I.P. on day 14 post inoculation. Tumors were excised 24, 48 and 96 h post injection (**A**). Representative flow plots showing changes in the Live/CD45^+^ IFN-γ^+^ population following GEM administration. All frequencies shown are of the parent gate. (**B**). Change in Live/CD45^+^ IFN-γ^+^ cell number. (**C**). Change in the proportion of Live/CD45^+^ cells that are IFN-γ^+^. (**D**). Change in Geometric Mean Fluorescence Intensity (GMF) of IFN-γ on Live/CD45^+^ IFN-γ^+^ cells. (**E**). Changes in the proportion of IFN-γ^+^ cells that are CD8α^+^ T cells (**Ei**), the proportion of CD8α^+^ T cells that are IFN-γ^+^ (**Eii**), and the GMF of IFN-γ on IFN-γ^+^ CD8α^+^ T cells (**Eiii**). (**F**). Changes in the proportion of IFN-γ^+^ cells that are CD4^+^ helper T cells (**Fi**), the proportion of CD4^+^ helper T cells that are IFN-γ^+^ (**Fii**), and the GMF of IFN-γ on IFN-γ^+^ CD4^+^ helper T cells (**Fiii**). (**G**). Changes in the proportion of IFN-γ^+^ cells that are granulocytes (**Gi**), the proportion of granulocytes that are IFN-γ^+^ (**Gii**), and the GMF of IFN-γ on IFN-γ^+^ granulocytes (**Giii**). (n = 8) (2way ANOVA with multiple comparisons: * *p* < 0.05, ** *p* < 0.01, *** *p* < 0.001; ROUT Outliers analysis with Q = 0.1%). All points represent mean ± SD.

**Figure 5 cells-13-01317-f005:**
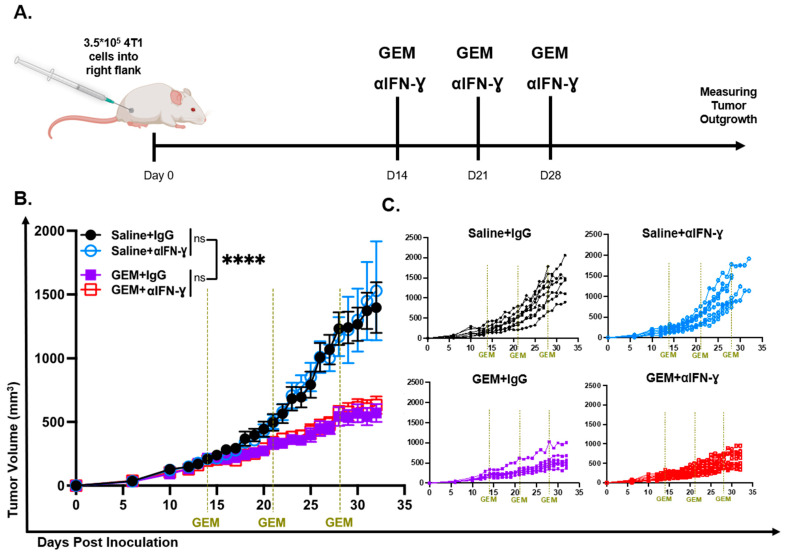
GEM-mediated tumor growth control is not dependent on IFN-γ. (**A**). Overview of experimental design to evaluate the efficacy of GEM on constraining TNBC primary tumor outgrowth when neutralizing IFN-γ. (**B**). Average tumor growth curves for mice treated with GEM (1.2 mg) or saline I.P. on days 14, 21 and 28 post inoculation + anti-IFN-γ (XMG1.2; 200 μg) or + IgG isotype control (200 μg). (Mixed-effects model: ns = non-significant, **** *p* < 0.0001). All points represent mean ± SEM. (**C**). Individual tumor growth curves of GEM and saline + anti-IFN-γ (XMG1.2) or + IgG isotype control treated mice.

**Figure 6 cells-13-01317-f006:**
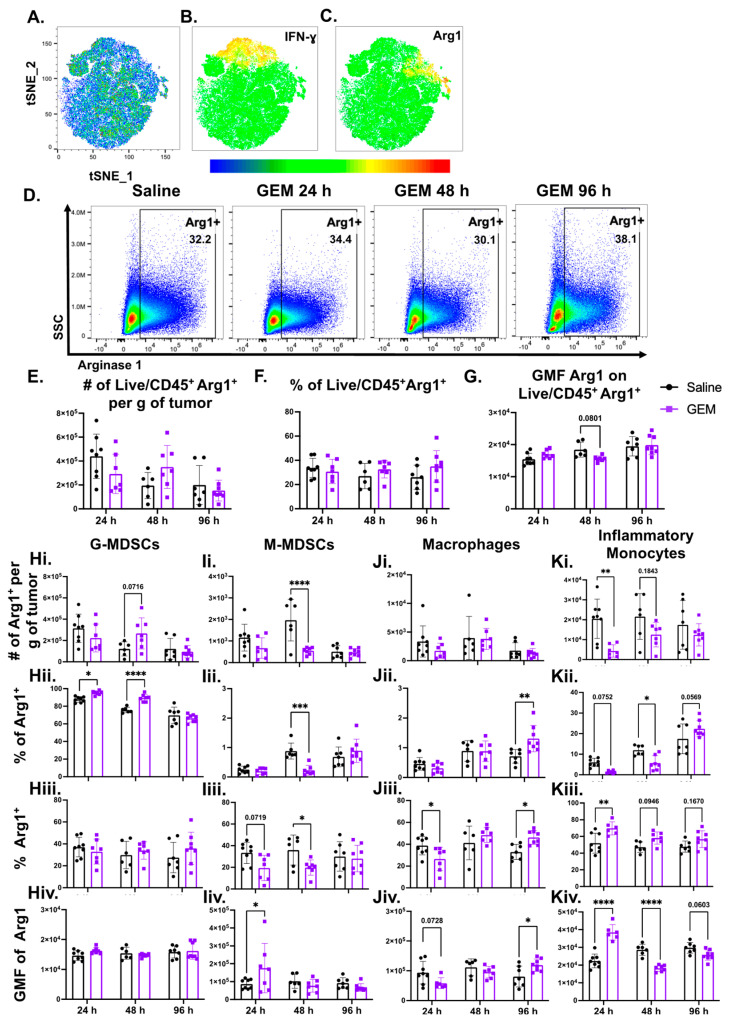
GEM treatment alters Arginase 1 (Arg1) expression by myeloid cells in TNBC tumors. 350 k 4T1 parental cells were inoculated in the right flanks of BALB/c mice. Mice were injected with GEM (1.2 mg) or saline I.P. on day 14 post inoculation. Tumors were excised 24, 48 and 96 h post injection (**A**). tSNE dimensionality reduction on granulocytic (Ly6G^+^CD11b^+^) cells in the TME. (**B**). Multigraph color mapping of IFN-γ expression on granulocytic cells in the TME, with green being lowly expressed and red being highly expressed. (**C**). Multigraph color mapping of Arg1 expression on granulocytic cells in the TME, with green being lowly expressed and red being highly expressed. (**D**). Representative flow plots showing changes in the Live/CD45^+^ Arg1^+^ population following GEM administration. All frequencies shown are of the parent gate. (**E**). Change in Live/CD45^+^ Arg1^+^ cell number. (**F**). Change in the proportion of Live/CD45^+^ cells that are Arg1^+^. (**G**). Change in Geometric Mean Fluorescence Intensity (GMF) of Arg1 on Live/CD45^+^ Arg1^+^ cells. (**H**). Changes in the number of Arg1^+^ G-MDSCs (Ly6G^+^CD11b^+^) (**Hi**), the proportion of Arg1^+^ cells that are G-MDSCs (**Hii**), the proportion of G-MDSCs that are Arg1^+^ (**Hiii**), and the GMF of Arg1 on Arg1^+^ G-MDSCs (**Hiv**). (**I**). Changes in the number of Arg1^+^ M-MDSCs (CD11b^+^Ly6G^−^F4/80^−^Ly6C^+^) (**Ii**)**,** the proportion of Arg1^+^ cells that are M-MDSCs (**Iii**), the proportion of M-MDSCs that are Arg1^+^ (**Iiii**), and the GMF of Arg1 on Arg1^+^ M-MDSCs (**Iiv**). (**J**). Changes in the number of Arg1^+^ macrophages (CD11b^+^Ly6G^−^F4/80^+^Ly6C^−^) (**Ji**), the proportion of Arg1^+^ cells that are macrophages (**Jii**), the proportion of macrophages that are Arg1^+^ (**Jiii**), and the GMF of Arg1 on Arg1^+^ macrophages (**Jiv**). (**K**). Changes in the number of Arg1^+^ inflammatory monocytes (CD11b^+^Ly6G^−^F4/80^+^Ly6C^hi^) (**Ki**), the proportion of Arg1^+^ cells that are inflammatory monocytes (**Kii**), the proportion of inflammatory monocytes that are Arg1^+^ (**Kiii**), and the GMF of Arg1 on Arg1^+^ inflammatory monocytes (**Kiv**). (n = 8) (2way ANOVA with multiple comparisons: * *p* < 0.05, ** *p* < 0.01, *** *p* < 0.001, **** *p* < 0.0001; ROUT Outliers analysis with Q = 0.1%). All points represent mean ± SD.

## Data Availability

Primary data can be made available for analysis upon reasonable request.

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
