# Peer review of "Tissue- and Temporal-Dependent Dynamics of Myeloablation in Response to Gemcitabine Chemotherapy"

_cells, 2024, doi:10.3390/cells13161317_

Round 1

Reviewer 1 Report

Comments and Suggestions for Authors

Article by Dr. Bullock and group heightening on the tissue-specific and Temporal Dynamics of Myeloablation  in Response to Gemcitabine Chemotherapy. This is a very well-written and well-planned experimental research article with a translational aspect. A few things need to be addressed before it is ready for acceptance. They are as follows:

1. In line 497, where authors discuss the gemcitabine resistance factors, they must add a line discussing the role of the NRF2 (antioxidant pathway) in gemcitabine resistance. It has been shown by several studies that NRF2 plays a significant role in creating resistance against chemotherapies, including Gemcitabine treatment (PMID: 31911550 and PMID: 37926068).

2. Authors should add a model depicting the global message of this article and discuss it in a few lines. 

3. Figure 3 must be edited. Texts inside the figures are not readable at all. Please increase the text sizes. 

Reviewer 2 Report

Comments and Suggestions for Authors

Triple-negative breast cancer (TNBC) due to a limited 5-year survival rate needs therapy improvement. Gemcitabine (GEM) is one of the chemotherapeutic options and has been reported to have favorable immunologic effects. In particular, GEM has been reported to be systemically myeloablative and can be considered a useful adjuvant therapy for promoting anti-tumor immunity.

The influence of GEM on tumor microenvironment (TME) was unknown. Thus, the presented study aimed to compare the impact of GEM on immune cells' presence and activity between TME and peripheral action.

The authors showed that treatment of murine TNBC with the clinically employed chemotherapy GEM results in tissue- and temporal-dependent immune ablative effects. GEM acts as a pan-myeloablative drug in the periphery, but in parallel it has an isolated impact on F4/80+Ly6Chi myeloid cells in the TME. A single dose of GEM resulted in a sustained reduction of T-lymphocytes in the TME; explaining why GEM as a monotherapy is not able to induce a strong adaptive immune response against TNBC. Interestingly, based on the study results Arginase 1 inhibitors should have beneficial effects when combined with GEM.

The results are convincing and bring new knowledge to the field of immunological response activation in breast cancer tumors. Data are generated in vivo, so the results might be soon evaluated in the more advanced study models.

There are small issues concerning the manuscript:

1. In the current version of the manuscript the right part of Figure 1 is cut. It must be corrected.

2. There are several typos along the text, e.g., in line 257 "weas" instead of "was". Please check the manuscript carefully.

3. Please check the length of the abstract. It should be up to 200 words and is approx. 215.

Comments on the Quality of English Language

There are several typos along the text, e.g., in line 257 "weas" instead of "was". Please check the manuscript carefully.

Round 2

Reviewer 1 Report

Comments and Suggestions for Authors

Accepted